# DAMamba: Vision State Space Model with Dynamic Adaptive Scan

**Tanzhe Li**[1,2,3,*]  **Caoshuo Li**[1,2,3,*]  **Jiayi Lyu**[4]  **Hongjuan Pei**[4]  **Baochang Zhang**[5],
**Taisong Jin**[1,2,3,†]  **Rongrong Ji**[1,2]

[1]Key Laboratory of Multimedia Trusted Perception and Effcient Computing,
Ministry of Education of China, Xiamen University, China
[2]School of Informatics, Xiamen University, China
[3]Fujian Ocean Innovation Center, China
[4]School of Engineering Science, University of Chinese Academy of Sciences, China
[5]School of Artificial Intelligence, Beihang University, China
{litanzhe, caoshuoli}@stu.xmu.edu.cn,jintaisong@xmu.edu.cn

## Abstract

State space models (SSMs) have recently garnered significant attention in computer vision. However, due to the unique characteristics of image data, adapting SSMs from natural language processing to computer vision has not outperformed the state-of-the-art convolutional neural networks (CNNs) and Vision Transformers (ViTs). Existing vision SSMs primarily leverage manually designed scans to flatten image patches into sequences locally or globally. This approach disrupts the original semantic spatial adjacency of the image and lacks flexibility, making it difficult to capture complex image structures. To address this limitation, we propose Dynamic Adaptive Scan (DAS), a data-driven method that adaptively allocates scanning orders and regions. This enables more flexible modeling capabilities while maintaining linear computational complexity and global modeling capacity. Based on DAS, we further propose the vision backbone DAMamba, which significantly outperforms popular vision Mamba models in vision tasks such as image classification, object detection, instance segmentation, and semantic segmentation. Notably, it surpasses some of the latest state-of-the-art CNNs and ViTs. Code is available at `https://github.com/ltzovo/DAMamba`.

## 1   Introduction

In recent years, to tackle the limitations of traditional convolutional neural networks (CNNs) [30] in modeling long-range dependencies, Transformer [51] have been introduced into computer vision, achieving state-of-the-art performance in image classification task. However, the self-attention mechanism within Transformer, due to its quadratic computational complexity, faces limitations when applied to high-resolution vision downstream tasks such as object detection and image segmentation. To address this issue, researchers have proposed various sparse attention mechanisms [54, 55, 58, 9, 69, 27, 67, 44]. These mechanisms reduce complexity by introducing sparsity into attention computations, but this usually comes at the expense of the model's global modeling capability, limiting their performance in some vision tasks.

State space models (SSMs) [13], represented by Mamba [12], have recently garnered significant attention from researchers. The core module, the S6 block, selectively retains or discards information

---

*Equal Contribution
†Corresponding Author

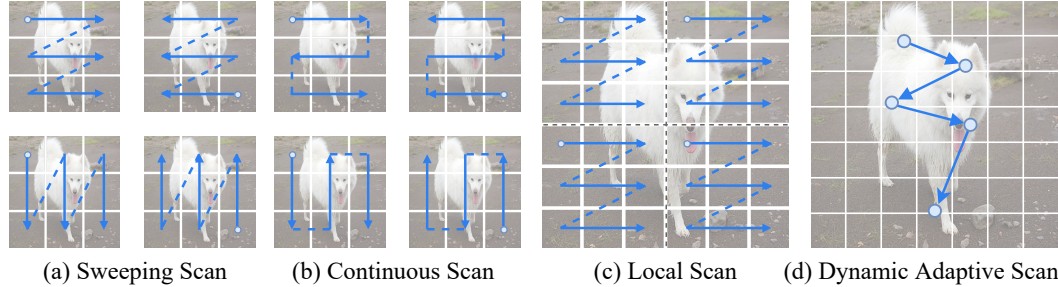

| (a) Sweeping Scan | (b) Continuous Scan | (c) Local Scan | (d) Dynamic Adaptive Scan |

Figure 1: Illustration of different scanning methods in vision state space models. As showed in Figure (a), (b), and (c): previous methods such as Vim [70], VMamba [35], PlainMamba [61], and LocalMamba [25] relies on manually designed global or local scanning methods. These fixed processing approaches lack flexibility and struggle to capture complex image structures. In the Figure (d), we propose a novel scanning method that adaptively allocates scanning order and regions through a data-driven approach. This not only achieves more flexible modeling capabilities but also maintains Mamba's linear computational complexity and global modeling capacity.

based on the relevance of each element in a sequence. By incorporating a selective mechanism for parallel computation alongside hardware-aware optimization, the S6 block not only offers a comprehensive global receptive field but also attains a computational complexity that scales linearly with the sequence length. This characteristic enables Mamba to outperform popular Transformer models in natural language processing tasks. Inspired by Mamba, some research efforts, such as Vim [70] and VMamba [35], have extended its application to computer vision. These approaches divide 2D images into patches and adopt specific scanning strategies to flatten the images into multiple 1D sequences from different directions. This successfully integrates the Mamba model into vision tasks, achieving promising performance and showcasing the potential of SSMs in computer vision.

Unlike one-dimensional sequential language data, visual data typically exhibits two-dimensional spatial structure. One of the core challenges in adapting the Mamba model for vision tasks lies in designing an appropriate scanning strategy for an image. Scanning strategy enables SSMs, which are designed for 1D sequence processing, to effectively accommodate the 2D spatial structure of images. Currently, scanning strategies for vision SSMs can be broadly categorized into three types: sweeping scan, continuous scan, and local scan. Vim and VMamba adopt the sweeping scan strategy, simulating a row-by-row scan from left to right and top to bottom, allowing Mamba to adapt to the 2D spatial structure of images. However, PlainMamba [61] argues that sweeping scan overlooks the importance of spatial continuity within images and thus introduces the continuous scan strategy to ensure the correlation between adjacent patches. Meanwhile, LocalMamba [25] proposes the local scan strategy, aiming to capture the local spatial relationships within images.

Although the aforementioned methods have proven effective in practice, they rely on manually designed scanning patterns that are independent of the input data, which may not be optimal. For instance, sweeping and continuous scan can cause spatially close patches to become distant in SSM computations, resulting in a loss of local information. On the other hand, local scan can capture local spatial relationships, limiting the model's ability to capture long-range dependencies. Clearly, there is a need for a more flexible scanning strategy that can dynamically adjust the scanning regions based on the characteristics of each input data instance. For example, in the case of an image of a dog, an ideal scan strategy should adaptively focus on the dog's body while filtering out irrelevant background information. However, such dynamic adjustment is beyond the capabilities of existing manually designed scanning approaches.

To address the aforementioned issues, we propose a flexible and efficient scanning strategy, named Dynamic Adaptive Scan (DAS). Unlike traditional manually designed scanning methods, DAS dynamically learns and adjusts scanning regions and their sequences during training, enabling smarter and more precise feature extraction. Specifically, DAS starts by defining a set of learnable positions, with initial values corresponding to the original locations of each patch. Then, through a learnable offset prediction network (OPN), a set of offset values is generated for each patch. By combining these offset values with the original patch positions, the predicted patch positions are computed.

Using bilinear interpolation, these predicted positions are gradient-linked to the feature map, allowing the offsets to be adaptively optimized during training. The predicted patches are arranged from top to bottom and left to right based on their original positions, dynamically forming a new sequence order according to the input data. Through this mechanism, DAS focuses on more critical regions, capturing important features and complex spatial structures with greater flexibility.

Based on the proposed DAS, we develop a powerful vision Mamba model, termed DAMamba. DAMamba can serve as a versatile vision backbone for various vision tasks. For instance, our DAMamba-T achieves an image classification accuracy of 83.8%, 48.5 $AP^b$ in object detection, 43.4 $AP^m$ in instance segmentation, and 50.3 mIoU in semantic segmentation. These results surpass the previous state-of-the-art Vision Mamba, VMamba, by 1.2% in classification accuracy, 1.2 $AP^b$, 0.7 $AP^m$, and 2.3 mIoU. Moreover, DAMamba also outperforms some recent state-of-the-art ViTs and CNNs in these vision tasks, demonstrating its superior performance and various applicability.

## 2 Related Work

### 2.1 Vision State Space Models

Although the Transformer [51] has achieved remarkable success in natural language processing, its quadratic complexity poses challenges when handling long sequence structures. To address this issue, state-space models [13] (SSMs), represented by Mamba [12], have gradually emerged as an alternative to Transformers. In visual tasks, the quadratic complexity of the standard self-attention mechanism similarly presents challenges for processing high-resolution images. Thus, the Vim [70] and VMamba [35] attempt to incorporate Mamba into computer vision tasks. However, inputting images into SSM models remains a critical challenge. Vim and VMamba address this by employing bidirectional and four-directional scanning strategies to transform image patches into one-dimensional sequences. Building on this, subsequent research introduced continuous scanning [61] and local four-directional scan [25] to better align with the two-dimensional structure of images. Despite the significant achievements of Mamba models in computer vision, existing scanning methods heavily rely on manual design, making it difficult to dynamically and flexibly adapt to input variations. This limitation hinders the model's ability to capture complex two-dimensional structures. Therefore, our goal is to propose a vision Mamba model capable of adaptively and flexibly adjusting scanning paths based on input image, further enhancing its performance in vision tasks.

### 2.2 Vision Transformers

The Transformer [51] model was first introduced in 2017 for natural language processing (NLP) tasks. With its powerful global modeling capabilities and excellent parallelism, the Transformer quickly gained popularity in the NLP. By the end of 2020, Vision Transformer [10] (ViT) successfully extended the Transformer model to large-scale image classification tasks, achieving state-of-the-art performance. Subsequently, DeiT [49] improved ViT by introducing knowledge distillation [20] and more efficient training strategies, enabling effective training even on relatively small datasets such as ImageNet-1K [43]. Following this development trajectory, researchers proposed numerous hierarchical Transformer models that reduce computational complexity for high-resolution images through various sparse attention mechanisms. Notable examples include the Swin Transformer [36] and PVT [54, 55]. Subsequent research [54, 55, 58, 9, 69, 27, 67, 44] introduced various sparse attention mechanisms to strike a balance between global modeling capability and computational complexity. However, the global modeling capabilities of these improved sparse attention mechanisms still fall short of the standard self-attention mechanism.

### 2.3 Convolutional Neural Networks

Convolutional Neural Network (CNN) [30] was initially proposed for handwritten digit recognition, but it wasn't until the introduction of AlexNet [29] in 2012, which triggered the "ImageNet moment," that the full potential of CNNs was realized. This breakthrough led to a rapid development in computer vision, driven by the resurgence of neural networks, with CNNs becoming the standard architecture for computer vision tasks. During this period, many representative CNN models emerged, such as VGG [45], GoogLeNet [46], ResNet [18], DenseNet [23], DCN [66, 53], and EfficientNet [48]. These models focused on different aspects, including accuracy, efficiency, and scalability, while promoting

valuable design principles. In recent years, inspired by ViTs, some CNNs [37, 57, 7, 34] have incorporated large kernel convolutions to capture long-range dependencies, achieving performance competitive with ViT. At the same time, CNNs have been widely integrated into various ViTs and vision Mambas to enhance local modeling capabilities, creating a complementary synergy between the two approaches. These advancements have driven the diversification and convergence of model design in vision tasks.

# 3 Methodology

## 3.1 Preliminaries

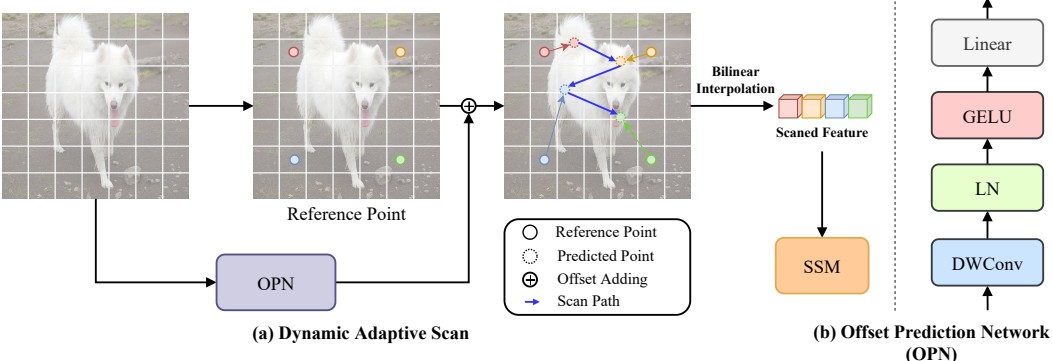

Figure 2: Illustration of the proposed Dynamic Adaptive Scan (DAS). For clarity, only four reference points are shown. **Left**: each initial reference point represents the original position of a patch, with its offsets learned by an Offset Prediction Network (OPN). Features of important regions are sampled based on the predicted 2D coordinates using bilinear interpolation. **Right** the detailed structure of the OPN is revealed. The query feature map is first transformed through depthwise convolution [22, 3] to integrate local information. Then, another linear layer, after layer normalization [1] and GELU [19] activation, converts the feature map into offset values.

State Space Models (SSMs) [13, 12] are a class of sequence modeling methods commonly used in deep learning, capable of representing dynamic systems through an intermediate latent state $h(t) \in \mathbb{R}^N$. Their core equations are as follows:

$$h'(t) = Ah(t) + Bx(t), \quad y(t) = Ch(t),$$

where the system matrices $A \in \mathbb{R}^{N \times N}$, $B \in \mathbb{R}^{N \times 1}$, and $C \in \mathbb{R}^{1 \times N}$ govern the dynamic evolution and output mapping.

To implement continuous-time models in practice, discretization techniques are required. The commonly used Zero-Order Hold (ZOH) method keeps the input constant within each time interval, transforming the continuous-time parameters $(A, B)$ into their discrete forms as follows:

$$\overline{A} = e^{\Delta A}, \quad \overline{B} = (\Delta A)^{-1}(e^{\Delta A} - I)\Delta B,$$

where $\Delta$ represents the sampling time scale. The resulting discretized model can then be expressed as:

$$h_t = \overline{A}h_{t-1} + \overline{B}x_t, \quad y_t = Ch_t.$$

This method not only supports efficient parallel computation but also directly generates sequence outputs through convolution operations:

$$y = x * \overline{K}, \quad \overline{K} = (C\overline{B}, C\overline{AB}, \dots, C\overline{A}^{L-1}\overline{B}),$$

where $\overline{K} \in \mathbb{R}^L$ is the SSM kernel, and $*$ denotes the convolution operation. This parallelization significantly enhances computational efficiency and scalability.

Although traditional SSMs (such as S4 [13]) achieve linear time complexity, their static parameterization limits their ability to capture the sequence context. To overcome this limitation, the Mamba [12]

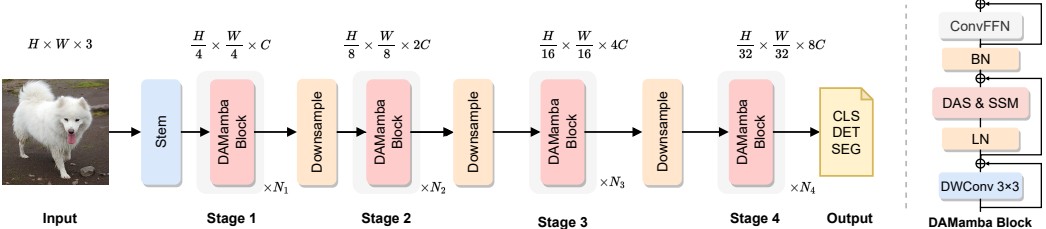

Figure 3: **Left**: The overall architecture of the proposed DAMamba, refer to Table 1 for configurations. **Right**: Details of an DAMamba Block.

introduces a dynamic and input-dependent parameterization mechanism. Unlike traditional models that use constant transition parameters $A, B$, the Mamba model dynamically computes the parameters $B \in \mathbb{R}^{B \times L \times N}$, $C \in \mathbb{R}^{B \times L \times N}$, and $\Delta \in \mathbb{R}^{B \times L \times D}$ from the input sequence $x \in \mathbb{R}^{B \times L \times D}$. This allows for richer and more sequence-aware dynamic modeling.

In Mamba, continuous-time parameters are adaptively adjusted to input-dependent parameters through selective functions based on the input sequence $x_t$:

$$\Delta_t = s_\Delta(x_t), \quad B_t = s_B(x_t), \quad C_t = s_C(x_t),$$

and the input-dependent discrete parameters $\overline{A_t}$ and $\overline{B_t}$ can be calculated accordingly. The discrete state transition and observation equations are as follows:

$$h_t = \overline{A_t}h_{t-1} + \overline{B_t}x_t, \quad y_t = C_t h_t.$$

The dynamic parameterization of the Mamba model not only improves sequence modeling performance but also demonstrates strong competitiveness in language modeling and vision tasks. For instance: Vim [70] combines bidirectional Mamba blocks to replace traditional Transformer blocks for visual modeling. VMamba [35] constructs a hierarchical structure by introducing 2D selective scanning, akin to the design of the Swin Transformer [36]. These advancements expand the application potential of SSMs in foundational vision tasks, further driving the development of SSM models.

## 3.2 Dynamic Adaptive Scan

As shown in Figure 2, we propose Dynamic Adaptive Scan (DAS), which effectively models the relationships between image patches under the guidance of important regions in the feature map. These focused regions are determined by multiple sets of learnable sampling points, which are learned by an offset prediction network (OPN) from the input image feature map. After obtaining the two-dimensional coordinates predicted by the OPN, we use bilinear interpolation to sample features from the feature map, and then input the sampled features into the SSM for feature aggregation. Furthermore, the positions of DAS provide stronger relative positional biases to facilitate the learning of the SSM.

Specifically, we first input the feature map into an OPN to predict the two-dimensional coordinate offsets $\Delta p \in \mathbb{R}^{H \times W \times 2}$ of the interested patches relative to the original patches:

$$\Delta p = OPN([x_1, x_2, \cdots, x_N]), \tag{1}$$

Then, these offsets are added to the positions of the original patches to determine the sampling locations of the interested patches:

$$p'[h, w, :] = p[h, w, :] + \Delta p[h, w, :], \tag{2}$$

where $p'$ and $p$ represent the sets of two-dimensional coordinates of the original patch and the interested patch on the feature map, respectively. $p'$ and $p$ take values between -1 and 1, (-1,-1) denotes the upper left corner, and (1,1) denotes the lower right corner. $h$ and $w$ denote the coordinates of the image patch in the height and width directions.

Next, we leverage position $p'$ for feature sampling, which is performed through a bilinear interpolation function. By establishing a relationship between the image patch features and the offsets, the proposed OPN can adaptively learn.

$$X'[h, w] = \phi(p'[h, w], X), \tag{3}$$

$$\phi(a,b) = \sum_{(r_x, r_y)} g(a_x, r_x, a_y, r_y) b[r_y, r_x, :], \tag{4}$$

$$g(c,d,e,f) = \max(0, 1 - |c - d|) \times \max(0, 1 - |e - f|), \tag{5}$$

where $g(c,d,e,f)$ represents the bi-linear sampling weight function, $a_x = p'[h,w,0]$, $a_y = p'[h,w,1]$, and $(r_x, r_y)$ indexes all locations on $X \in \mathbb{R}^{H \times W \times C}$. $g$ takes a nonzero value only at the four lattice points closest to the input location.

After obtaining the sampled feature vectors of interest, $X'$, we arrange them in the order of their original patches from top to bottom and left to right, and then input them into SSM for feature extraction.

### 3.3 Architecture Design of DAMamba

Table 1: Configurations for different DAMamba variants.

| Models | Channels | Blocks |
|---|---|---|
| DAMamba-F | [48, 96, 192, 256] | [2, 2, 10, 2] |
| DAMamba-T | [80, 160, 320, 512] | [3, 4, 12, 5] |
| DAMamba-S | [96, 192, 384, 512] | [4, 8, 20, 6] |
| DAMamba-B | [112, 224, 448, 640] | [4, 8, 25, 8] |

As shown in Figure 3, we propose a novel vision backbone named DAMamba base on DAS, and develop model variants at four scales: DAMamba-F, DAMamba-T, DAMamba-S and DAMamba-B. First, the input image $x \in \mathbb{R}^{H \times W \times 3}$ is processed through a stem module, consisting of 4 overlapping $3 \times 3$ convolutions, to generate a 2D feature map of size $\frac{H}{4} \times \frac{W}{4} \times C$. These features are then processed through four stages of continuous operations, forming a multi-scale hierarchical representation with resolutions of $\frac{H}{8} \times \frac{W}{8}$, $\frac{H}{16} \times \frac{W}{16}$, and $\frac{H}{32} \times \frac{W}{32}$, respectively. Each stage is composed of multiple stacked DAMamba blocks, with resolution progressively downsampled by a factor of 2 and the feature dimension increased. Additionally, the DAMamba architecture integrates convolutional positional encoding [4, 26, 31] and convolutional FFN (ConvFFN) [55], further enhancing the ability to extract local features. Throughout the architecture, DAMamba blocks are flexibly stacked across different stages to adapt to various downstream tasks. Finally, the model's output undergoes batch normalization, global average pooling, and a linear classification head to produce the final image classification task feature representation. The configurations for different model scales are provided in Table 1.

## 4 Experiments

### 4.1 Image Classification on ImageNet-1K

**Experimental settings:** We conducted image classification experiments based on the ImageNet-1K dataset [43], which consists of 1,281,167 training images and 50,000 validation images spanning 1,000 categories. The implementation of our experimental methods relied on the PyTorch [40] and Timm [56] libraries. To ensure a fair comparison, we adopted the commonly used experimental settings from DeiT [49]. The optimizer used was AdamW [38], with a cosine decay learning rate schedule and linear warm-up over the first 20 epochs. The models were trained for 300 epochs on images with a resolution of $224^2$. For data augmentation and regularization, we employed techniques such as RandAugmentation [6], Repeated Augmentation [21], Mixup [65], CutMix [64], Random Erasing [68], weight decay, label smoothing [47], and stochastic depth [24]. For the small and base versions of the model, we also used layerscale [50] and mesa [11] to help train better. During testing, center cropping was applied to the validation images to generate input images with a resolution of $224^2$. The experiments were conducted on 16 RTX 3090 GPUs. Notably, exponential moving average (EMA) [42] did not significantly improve the final model performance and was therefore not used in the experiments.

**Results:** As shown in Table 2, we compare the proposed DAMamba with several state-of-the-art models. The proposed DAMamba consistently outperforms ViT, CNN, and SSM models. Specifically,

Table 2: Results of DAMamba and the current state-of-the-art backbones on ImageNet-1K. All the models are trained and tested at $224 \times 224$ resolution.

| Model | Type | Params (M) | FLOPs (G) | Top-1 (%) |
|---|---|---|---|---|
| ConvNeXt V2-F [57] | CNNs | 5 | 0.8 | 78.0 |
| Vim-Ti [70] | SSMs | 7 | 1.5 | 76.1 |
| LocalVim-Ti [70] | SSMs | 8 | 1.5 | 76.2 |
| EffcientVMamba-T [41] | SSMs | 6 | 0.8 | 76.5 |
| DAMamba-F (ours) | SSMs | 6 | 1.3 | **79.1** |
| SLaK-T [34] | CNNs | 30 | 5.0 | 82.5 |
| ConvNeXt V2-T [57] | CNNs | 29 | 4.5 | 82.5 |
| InceptionNeXt-T [63] | CNNs | 28 | 4.2 | 82.3 |
| MambaOut-Tiny [62] | CNNs | 27 | 4.5 | 82.7 |
| UniRepLKNet-T [8] | CNNs | 31 | 4.9 | 83.2 |
| InternImage-T [53] | CNNs | 30 | 5.0 | 83.5 |
| Swin-T [36] | ViTs | 29 | 4.5 | 81.3 |
| CSwin-T [9] | ViTs | 23 | 4.3 | 82.7 |
| Agent-Swin-T [15] | ViTs | 29 | 4.5 | 82.6 |
| DAT-T [58] | ViTs | 29 | 4.6 | 82.0 |
| PVTv2-B2 [55] | ViTs | 26 | 4.0 | 82.0 |
| ClusterFormer-Tiny [32] | ViTs | 28 | - | 81.5 |
| Slide-PVT-S [39] | ViTs | 23 | 4.0 | 81.7 |
| NAT-T [16] | ViTs | 28 | 4.3 | 83.2 |
| $QFormer_h$-T [67] | ViTs | 29 | 4.6 | 82.5 |
| PartialFormer-B3 [52] | ViTs | 36 | 3.4 | 83.0 |
| StructViT-S-8-1 [28] | ViTs | 24 | 5.4 | 83.3 |
| Vim-S [70] | SSMs | 26 | 5.1 | 80.5 |
| VMamba-T [35] | SSMs | 30 | 4.9 | 82.6 |
| PlainMamba-L2 [61] | SSMs | 25 | 8.1 | 81.6 |
| LocalVMamba-T [25] | SSMs | 26 | 5.7 | 82.7 |
| EffcientVMamba-T [41] | SSMs | 33 | 4.0 | 81.8 |
| FractalMamba-T [59] | SSMs | 31 | 4.8 | 83.0 |
| DAMamba-T (ours) | SSMs | 26 | 4.8 | **83.8** |
| SLaK-S [34] | CNNs | 55 | 9.8 | 83.8 |
| InceptionNeXt-S [63] | CNNs | 49 | 8.4 | 83.5 |
| MambaOut-Small [62] | CNNs | 48 | 9.0 | 84.1 |
| UniRepLKNet-S [8] | CNNs | 56 | 9.1 | 83.9 |
| InternImage-S [53] | CNNs | 50 | 8.0 | 84.2 |
| Swin-S [36] | ViTs | 50 | 8.7 | 83.0 |
| Agent-Swin-S [15] | ViTs | 50 | 8.7 | 83.7 |
| NAT-S [16] | ViTs | 51 | 7.8 | 83.7 |
| PVTv2-B4 [55] | ViTs | 63 | 10.1 | 83.6 |
| DAT-S [58] | ViTs | 50 | 9.0 | 83.7 |
| ClusterFormer-Small [32] | ViTs | 49 | - | 83.4 |
| $QFormer_h$-S [67] | ViTs | 51 | 8.9 | 84.0 |
| BiFormer-B [69] | ViTs | 57 | 9.8 | 84.3 |
| PartialFormer-B4 [52] | ViTs | 64 | 6.8 | 83.9 |
| StructViT-B-8-1 [28] | ViTs | 52 | 12.0 | 84.3 |
| TransNeXt-Small [44] | ViTs | 50 | 10.3 | 84.7 |
| VMamba-S [35] | SSMs | 50 | 8.7 | 83.6 |
| PlainMamba-L3 [61] | SSMs | 50 | 14.4 | 82.3 |
| LocalVMamba-S [25] | SSMs | 50 | 11.4 | 83.7 |
| DAMamba-S (ours) | SSMs | 45 | 10.3 | **84.8** |
| ConvNeXt V2-B [57] | CNNs | 89 | 15.4 | 84.3 |
| SLaK-B [34] | CNNs | 95 | 17.1 | 84.0 |
| InceptionNeXt-B [63] | CNNs | 87 | 14.9 | 84.0 |
| MambaOut-Base [62] | CNNs | 85 | 15.8 | 84.2 |
| InternImage-B [53] | CNNs | 97 | 16.0 | 84.9 |
| Swin-B [36] | ViTs | 88 | 15.4 | 83.5 |
| CSwin-B [9] | ViTs | 78 | 15.0 | 84.2 |
| Agent-Swin-B [15] | ViTs | 88 | 15.4 | 84.0 |
| NAT-B [16] | ViTs | 90 | 13.7 | 84.3 |
| PVTv2-B5 [55] | ViTs | 82 | 11.8 | 83.8 |
| FLatten-Swin-B [14] | ViTs | 89 | 15.4 | 83.8 |
| DAT-B [58] | ViTs | 88 | 15.8 | 84.0 |
| $QFormer_h$-B [67] | ViTs | 90 | 15.7 | 84.1 |
| TransNeXt-Base [44] | ViTs | 90 | 18.4 | 84.8 |
| VMamba-B [35] | SSMs | 89 | 15.4 | 83.9 |
| DAMamba-B (ours) | SSMs | 86 | 16.3 | **85.2** |

DAMamba-B achieves an accuracy of 85.2%, which is 1.3% higher than the current state-of-the-art SSM model (VMamba-B). Compared to the state-of-the-art CNN (ConvNeXt V2) and ViT (TransNext), the proposed DAMamba-T shows a significant improvement in accuracy. Even when the model is scaled to approximately 50M, 30M and 6M parameters, DAMamba achieves top-1 accuracies of 84.8%, 83.8% and 79.1%, maintaining its excellent performance.

## 4.2 Object Detection and Instance Segmentation on COCO2017

**Experimental settings:** We conducted object detection and instance segmentation experiments on the COCO 2017 dataset. The COCO 2017 dataset [33] consists of approximately 118K training images and 5K validation images and serves as a commonly used benchmark for object detection

Table 3: Comparison of object detection and instance segmentation performance on COCO with Mask R-CNN detector. FLOPs are calculated with input resolution of $1280 \times 800$.

| Mask R-CNN 1× schedule | | | | | | | | |
|---|---|---|---|---|---|---|---|---|
| Backbone | $AP^b$ | $AP^b_{50}$ | $AP^b_{75}$ | $AP^m$ | $AP^m_{50}$ | $AP^m_{75}$ | #Param. | FLOPs |
| Swin-T | 42.7 | 65.2 | 46.8 | 39.3 | 62.2 | 42.2 | 48M | 267G |
| DAT-T | 44.4 | 67.6 | 48.5 | 42.4 | 66.1 | 45.5 | 48M | 272G |
| CSWin-T | 46.7 | 68.6 | 51.3 | 42.2 | 65.6 | 45.4 | 42M | 279G |
| ConvNeXt-T | 44.2 | 66.6 | 48.3 | 40.1 | 63.3 | 42.8 | 48M | 262G |
| PVTv2-B2 | 45.3 | 66.1 | 49.6 | 41.2 | 64.2 | 44.4 | 45M | 309G |
| QFormer$_h$-T | 45.9 | 68.5 | 50.3 | 41.5 | 65.2 | 44.6 | 49M | - |
| PartialFormer-B3 | 45.0 | - | - | 40.9 | - | - | 54M | 248G |
| BiFormer-S | 47.8 | 69.8 | 52.3 | 43.2 | 66.8 | 46.5 | - | - |
| MambaOut-T | 45.1 | 67.3 | 49.6 | 41.0 | 64.1 | 44.1 | 43M | 262G |
| VMamba-T | 47.3 | 69.3 | 52.0 | 42.7 | 66.4 | 45.9 | 50M | 271G |
| LocalVMamba-T | 46.7 | 68.7 | 50.8 | 42.2 | 65.7 | 45.5 | 45M | 291G |
| FractalMamba-T | 46.8 | 68.7 | 50.8 | 42.4 | 65.9 | 45.8 | 41M | 266G |
| DAMamba-T | **48.5** | **70.3** | **53.3** | **43.4** | **67.2** | **46.7** | 45M | 284G |
| Swin-S | 44.8 | 68.6 | 49.4 | 40.9 | 65.3 | 44.2 | 69M | 354G |
| Agent-Swin-S | 47.2 | 69.6 | 52.3 | 42.7 | 66.6 | 45.8 | - | 364G |
| DAT-S | 47.1 | 69.9 | 51.5 | 42.5 | 66.7 | 45.4 | 69M | 378G |
| CSWin-S | 47.9 | 70.1 | 52.6 | 43.2 | 67.1 | 46.2 | 54M | 342G |
| ConvNeXt-S | 45.4 | 67.9 | 50.0 | 41.8 | 65.2 | 45.1 | 70M | 348G |
| PVTv2-B3 | 47.0 | 68.1 | 51.7 | 42.5 | 65.2 | 45.7 | 63M | 397G |
| BiFormer-B | 48.6 | 70.5 | 53.8 | 43.7 | 67.6 | 47.1 | - | - |
| MambaOut-S | 47.4 | 69.1 | 52.4 | 42.7 | 66.1 | 46.2 | 65M | 354G |
| VMamba-S | 48.7 | 70.0 | 53.4 | 43.7 | 67.3 | 47.0 | 70M | 349G |
| LocalVMamba-S | 48.4 | 69.9 | 52.7 | 43.2 | 66.7 | 46.5 | 69M | 414G |
| DAMamba-S | **49.8** | **71.2** | **54.7** | **44.5** | **68.4** | **48.2** | 65M | 395G |
| Swin-B | 46.9 | 69.2 | 51.6 | 42.3 | 66.0 | 45.5 | 88M | 496G |
| CSwin-B | 48.7 | 70.4 | 53.9 | 43.9 | 67.8 | 47.3 | 88M | 496G |
| ConvNeXt-B | 47.0 | 69.4 | 51.7 | 42.7 | 66.3 | 46.0 | 107M | 486G |
| PVTv2-B5 | 47.4 | 68.6 | 51.9 | 42.5 | 65.7 | 46.0 | 102M | 557G |
| ViT-Adapter-B | 47.0 | 68.2 | 51.4 | 41.8 | 65.1 | 44.9 | 102M | 557G |
| MambaOut-B | 47.4 | 69.3 | 52.2 | 43.0 | 66.4 | 46.3 | 100M | 495G |
| VMamba-B | 49.2 | 71.4 | 54.0 | 44.1 | 68.3 | 47.7 | 108M | 485G |
| DAMamba-B | **50.6** | **71.9** | **55.5** | **44.9** | **68.9** | **48.7** | 105M | 520G |

and instance segmentation tasks. To evaluate the performance of the proposed model on downstream vision tasks, we selected DAMamba as the backbone network and embedded it into a detector to extract object and instance features from images.DAMamba was integrated into the classic Mask R-CNN [17] detector and initialized with weights pre-trained on the ImageNet-1K dataset for 300 epochs. For the object detection and instance segmentation tasks, we trained the model for 12 epochs (1×) and 36 epochs (3×). All experiments were conducted using the MMDetection [2] framework.

**Results:** The object detection and instance segmentation results of DAMamba on the COCO2017 dataset are shown in Table 6. In terms of bounding box and mask average precision ($AP^b$ and $AP^m$), DAMamba demonstrates outstanding performance. Using a (1×) fine-tuning schedule, DAMamba-T/S/B achieves object detection mAPs of 48.5/49.8/50.6, outperforming VMamba-T/S/B by 1.2/1.1/1.4 mAP, Swin-T/S/B by 5.8/5.0/3.7 mAP, and ConvNeXt-T/S/B by 4.3/4.4/3.6 mAP. Under the same configuration, the instance segmentation mAP of DAMamba-T/S/B also significant outperform VMamba-T/S/B, Swin-T/S/B and ConvNeXt-T/S/B.

## 4.3 Semantic Segmentation on ADE20K

**Experimental settings:** We conducted semantic segmentation experiments using the ADE20K dataset and performed a comparative analysis of DAMamba and other models within the UperNet [60] framework. In the UperNet framework, the backbone network was initialized with weights pre-trained on the ImageNet-1K dataset, while the remaining parts were randomly initialized. The model optimization employed the AdamW optimizer with a batch size of 16. To ensure a fair comparison, all models were trained for 160k iterations within the UperNet framework. All experiments were conducted using the MMSegmentation [5] framework.

**Results:** Table 4 presents the semantic segmentation results of DAMamba under the UperNet [60] framework. The experiments show that DAMamba-T/S/B achieves mIoU scores of 50.3%, 51.2%,

Table 4: Comparison of semantic segmentation on ADE20K with UPerNet segmentor. FLOPs are calculated with input resolution of $512 \times 2048$. 'SS' and 'MS' represent single-scale and multi-scale testing, respectively.

| Method | mIoU (SS) | mIoU (MS) | #Param. | FLOPs |
|---|---|---|---|---|
| UniRepLKNet-T | 48.6 | 49.1 | 61M | 946G |
| ConvNeXt-T | 46.0 | 46.7 | 60M | 939G |
| Swin-T | 44.4 | 45.8 | 60M | 945G |
| Agent-Swin-T | 46.7 | - | 61M | 954G |
| NAT-T | 47.1 | 48.4 | 58M | 934G |
| QFormer$_h$-T | 46.9 | 48.1 | 61M | - |
| PartialFormer-B3 | 47.0 | - | 65M | 923G |
| BiFormer-S | 49.8 | 50.8 | - | - |
| MambaOut-T | 47.4 | 48.6 | 54M | 938G |
| VMamba-T | 48.0 | 48.8 | 62M | 949G |
| LocalVMamba-T | 47.9 | 49.1 | 57M | 970G |
| FractalMamba-T | 48.0 | 48.9 | 53M | 942G |
| DAMamba-T | **50.3** | **51.2** | 55M | 937G |
| UniRepLKNet-S | 50.5 | 51.0 | 86M | 1036G |
| Swin-S | 47.6 | 49.5 | 81M | 1039G |
| Agent-Swin-S | 48.1 | - | 81M | 1043G |
| ConvNeXt-S | 48.7 | 49.6 | 82M | 1027G |
| NAT-S | 48.0 | 49.5 | 82M | 1010G |
| QFormer$_h$-S | 48.9 | 50.3 | 82M | - |
| PartialFormer-B3 | 48.3 | - | 95M | 1005G |
| BiFormer-B | 51.0 | 51.7 | - | - |
| MambaOut-S | 49.5 | 50.6 | 76M | 1032G |
| VMamba-S | 50.6 | 51.2 | 82M | 1028G |
| LocalVMamba-S | 50.0 | 51.0 | 81M | 1095G |
| DAMamba-S | **51.2** | **52.0** | 75M | 1050G |
| Swin-B | 48.1 | 49.7 | 121M | 1188G |
| Agent-Swin-B | 48.7 | - | 121M | 1196G |
| ConvNeXt-B | 49.1 | 49.9 | 122M | 1170G |
| NAT-B | 48.5 | 49.7 | 123M | 1137G |
| QFormer$_h$-B | 49.5 | 50.6 | 123M | - |
| MambaOut-B | 49.6 | 51.0 | 112M | 1178G |
| VMamba-B | 51.0 | 51.6 | 122M | 1170G |
| DAMamba-B | **51.9** | **52.3** | 117M | 1178G |

Table 5: Ablation studies on DAMamba-F for module designs.

| Module design | #Param. (M) | FLOPs (G) | Top-1 acc (%). |
|---|---|---|---|
| Baseline | 5.31M | 1.20G | 77.7 |
| + DAScan | 5.41M | 1.23G | 78.3 |
| + Convpos | 5.43M | 1.24G | 78.6 |
| + ConvFFN | 5.52M | 1.26G | 79.1 |

and 51.9%, respectively, significantly outperforming other types of models. The performance improvement of our DAMamaba is also evident when using multi-scale tests. These results further validate the exceptional generalization capability of DAMamba in downstream tasks.

## 4.4 Ablation Study

To validate the effectiveness of our method, we conducted image classification ablation experiments on ImageNet-1k using DAMamba-F in Table 5. Compared to the baseline model with Sweeping Scan, our proposed Dynamic Adaptive Scan (DAS) improves the top-1 accuracy by 0.6% while consuming only a small amount of additional FLOPs and parameter overhead. Furthermore, for the vision SSMs, which excels at global modeling, we observe that using convolutional positional encoding (Convpos) [4, 26, 31] and ConvFFN [55] for local modeling can improve accuracy by 0.3% and 0.5%, respectively.

# 5 Conclusion

In this paper, we have proposed a novel vision state space model, termed DAMamba. DAMamba significantly enhances the flexibility of modeling in vision SSMs and improves the ability to capture complex image structures, while maintaining both local and global contextual understanding. Specifically, a Dynamic Adaptive Scan mechanism is proposed to adaptively allocate scanning order and regions based on the input image. Extensive experiments on various datasets and popular vision tasks demonstrate that the proposed DAMamba significantly and consistently outperforms the state-of-the-art vision SSMs as well as popular ViT and CNN architectures, establishing new benchmarks for image classification, object detection, instance segmentation, and semantic segmentation. Our findings underscore the importance of the scanning mechanism in vision SSMs and highlight the tremendous potential of SSMs as vision backbone.

# 6 Acknowledgements

The work was Supported by the National Science Fund for Distinguished Young Scholars (No.62025603) and National Natural Science Foundation of China (Nos. 62320106007,62072386,62521007), supported by Fujian Provincial Natural Science Foundation of China (No. 2025J01003), Supported by Fujian Ocean Innovation Center (No. 25FV0CZJ07), supported by Yunnan Provincial Major S&T Special Plan Project (No. 202402AD080001), Beijing Natural Science Foundation (No. L254018), Joint Fund of the Science and Technology R&D Program of Henan (No. 235200810031), Jing-Jin-Ji Incorporation Project of the Natural Science Foundation of Hebei Province (No. H2024202009), supported by the Open Foundation of Key Lab of Oracle Information Processing of MOE (No. OIP2024E002).

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

# A  Technical Appendices and Supplementary Material

## A.1  Model Efficiency Comparison

As shown in Figure 4, it can be seen that under the same inference throughput or accuracy, the accuracy or inference throughput of the proposed DAMamba significantly outperforms the SSMs, ViTs and CNNs, indicating that the proposed DAMamba achieves **state-of-the-art performance and efficiency.**

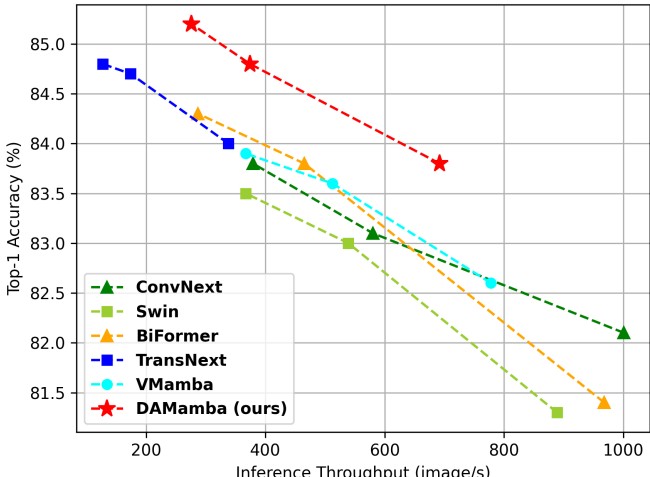

Figure 4: **The trade-off between ImageNet-1K top-1 accuracy and inference throughput.** All the models are trained under the DeiT training hyperparameters. The inference throughput is measured on an NVIDIA RTX 3090 GPU with a batch size 128.

## A.2  Visualization

As shown in Figure 5, we visualize the scanning results of DAMamba on ImageNet-1K images to verify the effectiveness of the proposed dynamic adaptive scanning method. For ease of visualization, we selected the final stage with fewer patches and removed prediction points outside the region of interest. For 2D positions of floating-point type, we visualize them by mapping each position to the nearest patch. We present three examples from the ImageNet-1K dataset. We observe that the proposed dynamic adaptive scanning primarily focuses on target objects, adapting to the input image by dynamically focusing on the foreground regions of interest. Additionally, the scanning areas can be adaptively adjusted.

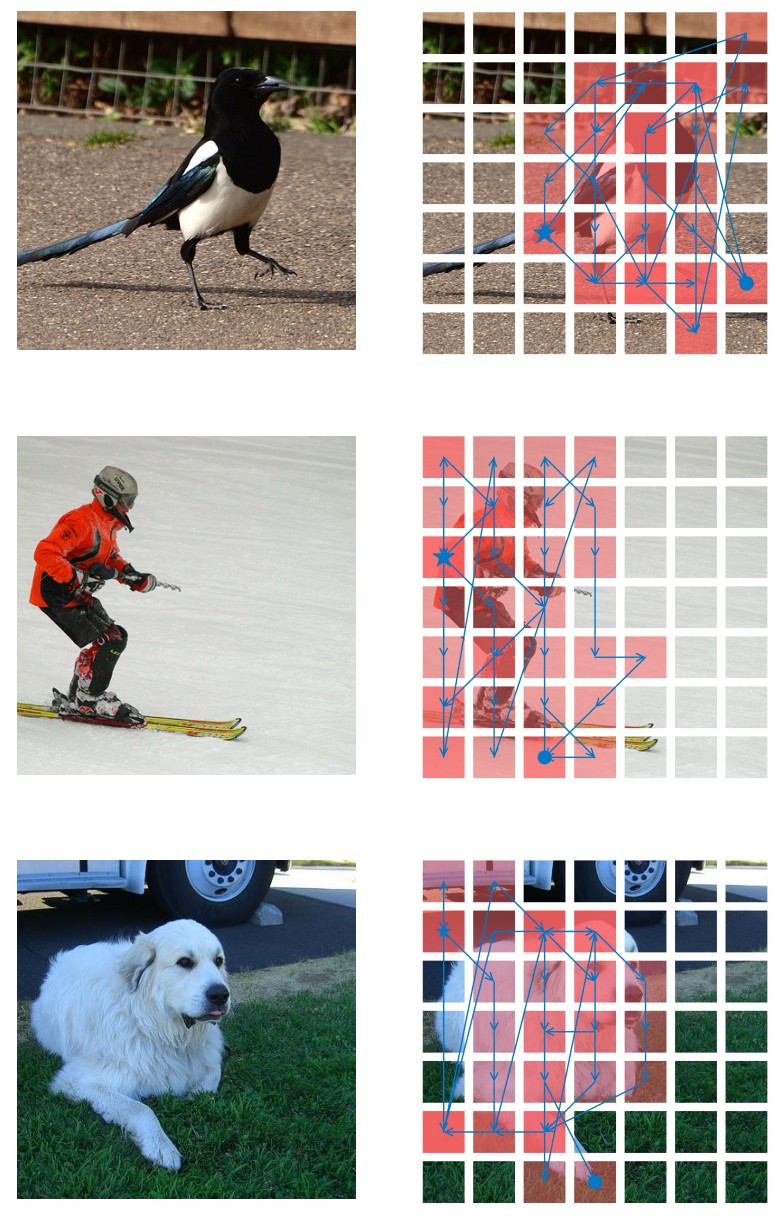

(a) Input image.                  (b) Scan Path.

Figure 5: Visualization of the Dynamic Adaptive Scan, where the blue pentagram represents the start of the scan and the blue circle represents the end of the scan.

## A.3 Object Detection and Instance Segmentatio with $3\times$ schedule

As shown in Table 6, when using a $3\times$ training configuration, DAMamba still maintains large performance improvements. These results indicate that DAMamba has the potential to achieve state-of-the-art performance in dense prediction downstream tasks.

Table 6: Comparison of object detection and instance segmentation performance on COCO with Mask R-CNN detector. FLOPs are calculated with input resolution of $1280 \times 800$.

| Backbone | $AP^b$ | $AP^b_{50}$ | $AP^b_{75}$ | $AP^m$ | $AP^m_{50}$ | $AP^m_{75}$ | #Param. | FLOPs |
|---|---|---|---|---|---|---|---|---|
| **Mask R-CNN 3× MS schedule** | | | | | | | | |
| Swin-T | 46.0 | 68.1 | 50.3 | 41.6 | 65.1 | 44.9 | 48M | 267G |
| PVTv2-B2 | 47.8 | 69.7 | 52.6 | 43.1 | 66.8 | 46.7 | 45M | 309G |
| ConvNeXt-T | 46.2 | 67.9 | 50.8 | 41.7 | 65.0 | 44.9 | 48M | 262G |
| NAT-T | 47.7 | 69.0 | 52.6 | 42.6 | 66.1 | 45.9 | 48M | 258G |
| QFormer$_h$-T | 47.5 | 69.6 | 52.1 | 42.7 | 66.4 | 46.1 | 49M | - |
| VMamba-T | 48.8 | 70.4 | 53.5 | 43.7 | 67.4 | 47.0 | 50M | 271G |
| LocalVMamba-T | 48.7 | 70.1 | 53.0 | 43.4 | 67.0 | 46.4 | 45M | 291G |
| DAMamba-T | **50.4** | **71.4** | **55.5** | **44.8** | **68.6** | **48.6** | 45M | 284G |
| Swin-S | 48.2 | 69.8 | 52.8 | 43.2 | 67.0 | 46.1 | 69M | 354G |
| PVTv2-B3 | 48.4 | 69.8 | 53.3 | 43.2 | 66.9 | 46.7 | 65M | 397G |
| ConvNeXt-S | 47.9 | 70.0 | 52.7 | 42.9 | 66.9 | 46.2 | 70M | 348G |
| NAT-S | 48.4 | 69.8 | 53.2 | 43.2 | 66.9 | 46.5 | 70M | 330G |
| QFormer$_h$-S | 49.5 | 71.2 | 54.2 | 44.2 | 68.3 | 47.6 | 70M | -G |
| VMamba-S | 49.9 | 70.9 | 54.7 | 44.2 | 68.2 | 47.7 | 70M | 349G |
| LocalVMamba-S | 49.9 | 70.5 | 54.4 | 44.1 | 67.8 | 47.4 | 69M | 414G |
| DAMamba-S | **51.2** | **72.1** | **56.1** | **45.1** | **69.2** | **49.1** | 65M | 395G |
| ConvNeXt-B | 48.5 | 70.1 | 53.3 | 43.5 | 67.1 | 46.7 | 108M | 486G |
| Swin-B | 48.6 | 70.0 | 53.4 | 43.3 | 67.1 | 46.7 | 107M | 496G |
| PVTv2-B5 | 48.4 | 69.2 | 52.9 | 42.9 | 66.6 | 46.2 | 102M | 557G |
| DAMamba-B | **51.4** | **72.3** | **56.4** | **45.3** | **69.5** | **48.9** | 105M | 520G |

