# OpenReview forum: "DAMamba: Vision State Space Model with Dynamic Adaptive Scan"
_NeurIPS.cc/2025/Conference — NeurIPS 2025 poster_

### Official Review · Reviewer_RoM8 · 2025-06-29

**Clarity:** 2
**Significance:** 3
**Originality:** 2
**Rating:** 4
**Confidence:** 5

**Summary:**

Existing visual state-space models typically scan image content using predefined paths, often breaking spatial relationships.
This work therefore produces a dynamic adaptive scanning method for adaptively learning scanning paths and regions.
The authors build a visual backbone network called DAMamba based on this, showing good performance.

**Questions:**

See weaknesses.

**Ethical Concerns:**

["NO or VERY MINOR ethics concerns only"]

**Final Justification:**

The authors have addressed my concerns in their responses.
Combined with the better model performance reported for their method, these improvements have led me to raise my overall score.

Nevertheless, the manuscript still suffers from an incomplete survey of related work, which is a weakness that regrettably pervades much of the current neural network design.
Specifically, the manuscript still neglects a substantial body of related work on
- deformable operations,
- dynamic network architectures,
- recent sequence-modeling methods in vision, and
- alternative scanning strategies within the Mamba family.

A more complete discussion would help reviewers and readers better appreciate the proposed method’s advantages and situate it against existing approaches.

**Limitations:**

The authors did not discuss the limitations of the work and the social implications.

I feel that the current setup of the ablation experiments has obvious shortcomings, as I mentioned in WEAKNESSES.
The scanning mechanism is the core of the paper.
But the authors focus only on the final performance of the model, but do not provide a sufficiently explicit baseline.
On the one hand the current baseline exhibits leading performance, on the other hand the authors neglect a full discussion with existing scanning strategies under the same baseline.
These are very important for the paper.
It is hoped that the authors can further complement and improve this work.

**Quality:**

3

**Strengths And Weaknesses:**

## Strengths

- The authors propose a data-driven information scanning mechanism that adaptively learns scanning paths and regions of an image within the SSM framework.
- Based on the proposed dynamic scanning strategy, the authors construct a dynamic visual SSM backbone network that exhibits better performance compared to other existing visual backbones.

## Weaknesses

1\. Dynamic adaptation schemes for information aggregation paths are common in the development of deep neural networks nowadays. Examples include Deformable Convolution [1-4], Kernel [5], Transformer [6,7]. The authors have neglected the relevance of dynamic design in these important works to this paper, and lacked the necessary analysis and comparisons.

2\. In the related work and background introduction, the authors have also neglected important elements such as Dynamic Neural Networks [8] which are closely related to this paper.

3\. As a visual model based on sequence modeling strategy, the development of other existing sequence modeling strategies such as [9-11] is also neglected in the discussion and analysis.

4\. In ablation experiments, the performance of ConvPos and ConvFFN can only be treated as a re-validation of the validity of the existing components, and cannot be used as a performance contribution point in the model design. So it seems somewhat unreasonable to treat them as additional independent components for ablation. It might be more appropriate to merge the whole into the baseline as a more standard baseline.

5\. For the BASELINE model in the ablation experiment:

5.1 The performance of ConvPos and ConvFFN can only be taken as a re-validation of the validity of the existing components, and cannot be used as a performance contribution point in the model design. So it seems unreasonable to ablate them as additional independent components. It might be more appropriate to merge the whole into the baseline as a more standardized baseline.

5.2 Why does the current baseline model perform so well? From the experimental results, it has surpassed the well-designed Vim-Ti, LocalVim-Ti and EffcientVMamba-T.

6\. The ablation experiment also lacks a comparison and discussion of the scanning strategies that have been proposed in the current Vision Mamba-related studies (e.g., those introduced in the reviews [12,13,14]).


REF:

1. Deformable Convolutional Networks, 2017
2. Deformable ConvNets v2: More Deformable, Better Results, 2019
3. InternImage: Exploring Large-Scale Vision Foundation Models with Deformable Convolutions, 2023
4. Efficient Deformable ConvNets: Rethinking Dynamic and Sparse Operator for Vision Applications, 2024
5. Deformable Kernels: Adapting Effective Receptive Fields for Object Deformation, 2020
6. Vision Transformer with Quadrangle Attention, 2023
7. DAT++: Spatially Dynamic Vision Transformer with Deformable Attention, 2023
8. Dynamic Neural Networks: A Survey, 2021
9. Eagle and Finch: RWKV with Matrix-Valued States and Dynamic Recurrence, 2024
10. Retentive Network: A Successor to Transformer for Large Language Models, 2023
11. xLSTM: Extended Long Short-Term Memory, 2024
12. Visual Mamba: A Survey and New Outlooks, 2024
13. A Survey on Visual Mamba, 2024
14. Vision mamba: A comprehensive survey and taxonomy, 2024

---

> ### Author Rebuttal · Authors · 2025-07-31
>
> We sincerely appreciate your valuable and encouraging comments on our technical contributions and performance.
>
> **Q1:** Dynamic adaptation schemes for information aggregation paths are common in the development of deep neural networks nowadays. Examples include Deformable Convolution [1-4], Kernel [5], Transformer [6,7]. The authors have neglected the relevance of dynamic design in these important works to this paper, and lacked the necessary analysis and comparisons.
>
> **R1:** We thank the reviewer for raising this point. Our paper primarily focuses on the differences and design of scan strategies within Vision SSM architectures, and therefore does not provide a systematic analysis of the connections with Deformable Convolution, Kernel methods, and Transformers.
>
> In fact, Dynamic Adaptive Scan (DAS) shares a similar core idea with these methods—that is, dynamically determining the feature extraction regions based on the input content—although the modeling mechanisms differ:
>
> Deformable Convolution, Kernel methods, and Transformers mainly learn dynamically aggregated receptive regions; in contrast, DAS reorganizes the sequence input at the scan level, learning not only the dynamic aggregation of regions but also their global ordering. This endows Vision SSMs with an enhanced ability to adapt to the 2D image structure.
>
> We will supplement the final version with relevant citations (e.g., [1–7]) and provide additional necessary analysis and comparisons.
>
>
> **Q2:** In the related work and background introduction, the authors have also neglected important elements such as Dynamic Neural Networks [8] which are closely related to this paper.
>
> **R2:** Thank you for the reminder. Works on Dynamic Neural Networks (DNN) [8] indeed share conceptual connections with our proposed DAS method, especially regarding structural dynamicity. We will include this survey paper and some of its referenced works in the related work section.
>
>
> **Q3:** As a visual model based on sequence modeling strategy, the development of other existing sequence modeling strategies such as [9-11] is also neglected in the discussion and analysis.
>
> **R3:** We acknowledge that the current related work section primarily focuses on Vision Transformer and Mamba’s vision adaptation approaches, and does not systematically cover emerging sequence modeling architectures such as RWKV [9], RetNet [10], and xLSTM [11]. These methods have shown strong performance in language tasks and are gradually being adapted for vision applications, making them worthy of attention. We will expand the background introduction on sequence modeling to include these recent advances in the final version.
>
>
>
> **Q4:** In ablation experiments, the performance of ConvPos and ConvFFN can only be treated as a re-validation of the validity of the existing components, and cannot be used as a performance contribution point in the model design. So it seems somewhat unreasonable to treat them as additional independent components for ablation. It might be more appropriate to merge the whole into the baseline as a more standard baseline.
>
> **R4:** We agree with the reviewer’s opinion that ConvPos and ConvFFN are standard modules in popular vision models. Indeed, we should incorporate them into the “standard baseline configuration” for ablation studies. We will redo the ablation experiments accordingly in the final version as per your suggestion.
>
>
> **Q5:** Why does the current baseline model perform so well? From the experimental results, it has surpassed the well-designed Vim-Ti, LocalVim-Ti and EffcientVMamba-T.
>
> **R5:** This is because our DAMamba adopts a pyramidal hierarchical structure, whereas LocalVim and Vim use an isotropic structure (LocalVMamba and VMamba adopt a pyramidal hierarchical structure). In vision backbones, pyramidal structures typically outperform isotropic structures. For example, the pyramidal VMamba outperforms the isotropic Vim. Additionally, although EfficientVMamba-T also uses a pyramidal structure, under the same parameter size (6M), the baseline used in our DAMamba-P ablation study consumes more FLOPs (1.2G vs. 0.8G), which leads to better performance.
>
> **Q6:** The ablation experiment also lacks a comparison and discussion of the scanning strategies that have been proposed in the current Vision Mamba-related studies (e.g., those introduced in the reviews [12,13,14]).
>
> **R6:** To ensure a fair comparison, we did not use ConvPos and ConvFFN, taking DAMamba-P as the baseline and replacing its scan method with other scanning strategies for comparison. As shown below, our method clearly outperforms these alternatives. Due to limited time during the rebuttal, we only compared a few scanning strategies, but we will include more comparisons and discussions of scanning methods from surveys [12], [13], and [14] in the final version.
>
> Method  |Accuracy (%)|
> ------  | ------ |
> Dynamic Adaptive Scan(Ours)	|  78.3|
> Sweeping Scan (VMamba)|  77.7|
> Local Scan (LocalMamba) | 77.9|
> Continuous 2D Scan (Plainmamba) | 77.6 |
> Efficient 2D Scan(Efficientvmamba) | 77.3 |
>
>
> REF:
>
> 1.Deformable Convolutional Networks, 2017
>
> 2.Deformable ConvNets v2: More Deformable, Better Results, 2019
>
> 3.InternImage: Exploring Large-Scale Vision Foundation Models with Deformable Convolutions, 2023
>
> 4.Efficient Deformable ConvNets: Rethinking Dynamic and Sparse Operator for Vision Applications, 2024
>
> 5.Deformable Kernels: Adapting Effective Receptive Fields for Object Deformation, 2020
>
> 6.Vision Transformer with Quadrangle Attention, 2023
>
> 7.DAT++: Spatially Dynamic Vision Transformer with Deformable Attention, 2023
>
> 8.Dynamic Neural Networks: A Survey, 2021
>
> 9.Eagle and Finch: RWKV with Matrix-Valued States and Dynamic Recurrence, 2024
>
> 10.Retentive Network: A Successor to Transformer for Large Language Models, 2023
>
> 11.xLSTM: Extended Long Short-Term Memory, 2024
>
> 12.Visual Mamba: A Survey and New Outlooks, 2024
>
> 13.A Survey on Visual Mamba, 2024
>
> 14.Vision mamba: A comprehensive survey and taxonomy, 2024

---

> ### Comment · Reviewer_RoM8 · 2025-08-09
>
> The authors have addressed my concerns in their responses. Combined with the better model performance reported for their method, these improvements have led me to raise my overall score.
>
> Nevertheless, the manuscript still suffers from an incomplete survey of related work, which is a weakness that regrettably pervades much of the current neural network design. Specifically, the manuscript still neglects a substantial body of related work on
> - deformable operations,
> - dynamic network architectures,
> - recent sequence-modeling methods in vision, and
> - alternative scanning strategies within the Mamba family.
>
> A more complete discussion would help reviewers and readers better appreciate the proposed method’s advantages and situate it against existing approaches.

---

### Official Review · Reviewer_vgjh · 2025-06-29

**Clarity:** 2
**Significance:** 3
**Originality:** 3
**Rating:** 5
**Confidence:** 2

**Summary:**

This paper introduces a dynamic scanning order for the vision mamba architecture, which needs to process an image in a sequence and cannot process it in parallel. They show that prior work was limited in their human defined scanning orders, and introduce a scanning order that can be learned and is therefore data-dependent. They show state-of-the-art performance on image classification.

**Questions:**

- It is unclear how the initial points are chosen, are they learned?
- It is unclear why multiple points are derived from a single sample point and why they thus need to be ordered from left to right and top to bottom?

**Ethical Concerns:**

["NO or VERY MINOR ethics concerns only"]

**Final Justification:**

My concerns have mostly been addressed in the discussion period, see below.

**Limitations:**

- The method is not pure Mamba, but rather a hybrid of a CNN and Mamba.

**Paper Formatting Concerns:**

- Grammar can be improved, but nothing major.

**Quality:**

2

**Strengths And Weaknesses:**

Strengths:
- Scanning in a data-dependent order makes sense, this should be more effective than any human defined order.
- The classification results are strong and indeed seem to be state-of-the-art for the models tested.
- Visualisations of the adaptive scan clearly show that the scan focuses on the object of interest, confirming the efficacy of the method.

Weaknesses:
- The equations are improperly introduced/explained for anyone not familiar with how SSM's work.
- The method section is insufficient, it is unclear how certain things work (see Questions).
- The detection and instance/semantic segmentation results do not include the best competitors from the classification results (like TransNext). Looking at the TransNext paper for example, they seem to get higher performance for these tasks. Unless there is a good reason to leave them out, it should be clearly noted that on dense downstream tasks the proposed method is not state-of-the-art.
- Although the papers mostly centers around the dynamic scan, the convolutional inductive biases seem to be more impactful in the ablation study (+0.8) than the dynamic scan (+0.6).

---

> ### Author Rebuttal · Authors · 2025-07-31
>
> We sincerely appreciate your valuable and encouraging comments on our technical contributions and performance.
>
> **Q1:** It is unclear how the initial points are chosen, are they learned?
>
> **R1:** The reference points are not learned; they are predefined. Specifically, each patch in the image serves as an initial point, with its center position taken as the initial coordinate.
>
>
> **Q2:** It is unclear why multiple points are derived from a single sample point and why they thus need to be ordered from left to right and top to bottom?
>
> **R2:** Each reference point has only one offset point because scanning from top to bottom and left to right better aligns with human intuitive vision. Sorting the points according to their original coordinates—from top to bottom and left to right—helps preserve local adjacency and enhances semantic continuity. Moreover, the top-to-bottom, left-to-right scanning order is also a commonly used, simple, and effective approach in Vision Mamba models.
>
> **Q3:** The equations are improperly introduced/explained for anyone not familiar with how SSM's work.
>
> **R3:** We appreciate this feedback. Currently, the method section indeed assumes that readers have some background knowledge of State Space Models (SSM). To improve readability, we will, in the final version:
>
> 1.Provide a more detailed introduction and explanation of the SSM working principles;
> 2. Include pseudocode or clearer module descriptions covering offset generation, offset point feature computation, and sequence rearrangement.
>
> **Q4:** The method section is insufficient, it is unclear how certain things work (see Questions).
>
> **R4:** Thank you for your suggestion. In the final version, we will further improve the methodology section, especially by clarifying the data flow for initial point selection, offset calculation, offset point sampling, sorting, and aggregation processes. We will also introduce pseudocode and module diagrams for key components to reduce abstraction and enhance clarity.
>
>
> **Q5:** The detection and instance/semantic segmentation results do not include the best competitors from the classification results (like TransNext). Looking at the TransNext paper for example, they seem to get higher performance for these tasks. Unless there is a good reason to leave them out, .it should be clearly noted that on dense downstream tasks the proposed method is not state-of-the-art.
>
>
> **R5:** Thank you for your suggestion. Although the TransNext paper appears to achieve higher performance on these tasks, our method attains better accuracy in image classification. Additionally, for models of comparable size such as DAMamba-T, our inference speed is nearly twice that of TransNext-T. We will clearly state in the final version that our proposed method is not state-of-the-art on dense downstream tasks.
>
> **Q6:** Although the papers mostly centers around the dynamic scan, the convolutional inductive biases seem to be more impactful in the ablation study (+0.8) than the dynamic scan (+0.6).
>
> **R6:** Yes, for SSM models that excel at global modeling, convolutional local modeling is very important—especially for vision tasks where capturing local details is crucial.

---

> > ### Comment · Reviewer_vgjh · 2025-08-05
> >
> > Thanks to the authors for their rebuttal. The authors indicate that TransNext performs better on dense tasks, while worse on global classification. Could they provide an explanation why that is? Also, the authors mention their method is faster than TransNext, however inference speed is not reported in the paper. Based on the efficiency misnomer by Deghani et al, it might be deceiving to solely rely on parameters and flops. Therefore, it would help if the authors could provide FPS, so that it becomes clear that their method still provides a better trade off between speed and accuracy on dense tasks compared to prior methods. Furthermore, it would be helpful to see the impact on FPS for the added convolutional inductive biases, as these make the model no longer purely SSM based and might therefore slow it down.

---

> ### Author Response · Authors · 2025-08-06
>
> We sincerely thank the reviewer for the constructive comments and positive interaction. Below are our detailed responses to each question.
>
> **Q1:** The authors indicate that TransNext performs better on dense tasks, while worse on global classification. Could they provide an explanation why that is?
>
> **R1:** This is because TransNext uses significantly more FLOPs for dense vision tasks on high-resolution images. As shown in the table below, we tested the FLOPs of TransNext-T in object detection and instance segmentation tasks under the Mask R-CNN framework, with all models evaluated at a resolution of 1280×800. The results show that, given the same number of parameters, TransNext-T has far higher FLOPs than our DAMamba-T. Therefore, the original TransNext paper reported only the parameter counts for vision downstream tasks without providing FLOPs.
>
> Method | #Params(M)|FLOPs(G)|
> ------  | :-------------------:|  :-------------------: |
> DAMamba-T	|  45|284 |
> TransNext-T	|  48|371 |
>
>
>
> **Q2:** Also, the authors mention their method is faster than TransNext, however inference speed is not reported in the paper. Based on the efficiency misnomer by Deghani et al, it might be deceiving to solely rely on parameters and flops. Therefore, it would help if the authors could provide FPS, so that it becomes clear that their method still provides a better trade off between speed and accuracy on dense tasks compared to prior methods.
>
> **R2:** In **Section A.1 Model Efficiency Comparison (line 425 of the paper)**, we report a trade-off comparison between inference throughput and performance with other popular models. The results show that our model achieves the best balance between performance and speed.
>
> **Q3:** Furthermore, it would be helpful to see the impact on FPS for the added convolutional inductive biases, as these make the model no longer purely SSM based and might therefore slow it down.
>
> **R3:** Below are the ablation results of inference throughput for ConvPos and ConvFFN. It can be seen that adding convolutional inductive biases leads to a slight decrease in inference speed.。
>
> | Method        | Throughput (image/s) |
> |-------------|:-------------------:|
> | DAMamba-T      |         692         |
> | Remove Convpos|         727         |
> | Remove ConvFFN|         779         |

---

> > ### Comment · Reviewer_vgjh · 2025-08-06
> >
> > 1. Thanks for the respone. Why does TransNext-T use more FLOPs than DAMamba-T for dense tasks while similar FLOPs for classification?
> >
> > 2. Thanks for pointing me to figure 4. It might be helpful to move it into the main paper. Why is throughput used instead of inference time of single image?
> >
> > 3. Thanks for showing the impact on speed for these components. It may be also helpful to report this in the main paper.

---

> > > ### Author Response · Authors · 2025-08-07
> > >
> > > We greatly appreciate the reviewer’s valuable feedback and positive engagement. Below, we provide our detailed responses to each of the questions raised.
> > >
> > > **Q1:** Thanks for the respone. Why does TransNext-T use more FLOPs than DAMamba-T for dense tasks while similar FLOPs for classification?
> > >
> > > **R1:** TransNext adopts two mechanisms: sliding window attention and pooling attention. The computational complexity of the attention score and weighted summation parts in the pooling attention is:
> > > \begin{equation}
> > > \Omega ({\rm PA}) = 2(HW)\left(\frac{HW}{R^2}\right)C ,
> > > \end{equation}
> > >
> > > Here, \$H\$ and \$W\$ represent the height and width of the feature map, respectively, \$C\$ is the feature dimension, and \$R\$ is the size of the pooling window. In TransNext, \$R = 7\$. In image classification tasks, using a \$7 \times 7\$ pooling window strikes a good balance between computational complexity and performance.
> > >
> > > However, in downstream dense vision tasks, the input image resolution is typically much higher than in image classification. For example, image classification commonly uses an input size of 224×224, whereas object detection tasks often use a resolution of 1280×800 when evaluating FLOPs. To maintain performance, TransNext still adopts a fixed 7×7 pooling window for these high-resolution tasks, without adjusting the pooling window size according to the increased image resolution. This leads to a quadratic increase in computational complexity.
> > >
> > > As a result, compared to models like our DAMamba that maintain linear computational complexity, TransNext exhibits significantly higher FLOPs in downstream dense vision tasks.
> > >
> > >
> > > **Q2:** Thanks for pointing me to figure 4. It might be helpful to move it into the main paper. Why is throughput used instead of inference time of single image?
> > >
> > > **R2:** Thank you for your suggestion. Due to space limitations, we previously placed this section in the appendix; however, we will move it to the main text in the final version to enhance the completeness of the paper.
> > >
> > > The reason we use throughput as the evaluation metric is that many popular vision backbone networks, such as Swin Transformer and ConvNeXt, commonly use this metric for speed testing. Therefore, to ensure a fair comparison and maintain consistency with existing work, we chose the same measurement.
> > >
> > > **Q3:** Thanks for showing the impact on speed for these components. It may be also helpful to report this in the main paper.
> > >
> > > **R3:** Thank you for the helpful suggestion. We will include these results in the final version of the paper to help readers better understand the impact of different components on inference speed.

---

> > > > ### Comment · Reviewer_vgjh · 2025-08-08
> > > >
> > > > I thank the authors for their elaboration. I have no further questions and will raise my score.

---

> > > > > ### Author Response · Authors · 2025-08-08
> > > > >
> > > > > We sincerely thank the reviewer for the positive feedback and for considering raising the score.

---

### Official Review · Reviewer_xcKP · 2025-07-03

**Clarity:** 3
**Significance:** 2
**Originality:** 2
**Rating:** 5
**Confidence:** 5

**Summary:**

This paper investigates the adaptation of State Space Models (SSMs) for use as backbone architectures in computer vision tasks. The authors identify limitations with previous SSM approaches, especially the inflexibility and inefficiency of manually-designed image patch scanning schemes. To address these issues, they propose Dynamic Adaptive Scan (DAS), a data-driven method that learns to allocate scanning order and regions adaptively based on each input image. The Dynamic Adaptive Scan is integrated into a new vision SSM backbone, DAMamba, which demonstrates strong empirical results across classification, detection, instance segmentation, and semantic segmentation on standard benchmarks, consistently outperforming prior visual SSMs (e.g., VMamba) and in several cases surpassing popular CNN and Vision Transformer architectures.

**Questions:**

please see weaknesses.

**Ethical Concerns:**

["NO or VERY MINOR ethics concerns only"]

**Final Justification:**

The author rebuttal has addressed my concerns.

**Limitations:**

Yes.

**Paper Formatting Concerns:**

I have not noticed

**Quality:**

2

**Strengths And Weaknesses:**

Strengths

1. DAMamba is thoroughly evaluated on multiple standard benchmarks, including ImageNet-1K for classification, COCO2017 for object detection/segmentation, and ADE20K for semantic segmentation. In each case, DAMamba variants outperform comparable SSM baselines (e.g., VMamba, Vim) and are competitive with, or superior to, many top CNN and ViT architectures at similar parameter/FLOP budgets.

2. By providing different model scales and integrating with widely-used frameworks, the work has practical value and reproducibility.

Weaknesses

1. Limited Theoretical Discussion: The paper’s justification for DAMamba’s improved spatial modeling remains primarily empirical. While the practical benefits of DAS are evident, the work does not rigorously dissect or quantify how much spatial adjacency (or other higher-order image structure) is preserved versus prior scan methods, beyond anecdotal visualizations and gains in accuracy—this limits mechanistic insight.

2. I believe that DAMamba’s performance may stem from overfitting to a fixed resolution. I’m genuinely curious—what would happen if the authors extended DAMamba to different resolutions? If it fails to maintain strong performance across varying resolutions, then I would consider the overall architectural design to lack sufficient insight.

3. While major design elements are described, there are some notational ambiguities—e.g., in the formalism in Section 3.2 around offset prediction and feature sampling, more explicit pseudocode or implementation details would aid reproducibility for non-experts.

4. While Figure 5 in the supplement visually contrasts regions of attention, the main methods for inspecting what is actually learned by the scanning process are limited to qualitative renderings, lacking systematic quantitative evaluation (e.g., are certain classes of object consistently mis-scanned?)

---

> ### Author Rebuttal · Authors · 2025-07-31
>
> We sincerely appreciate your valuable and encouraging comments on our technical contributions and performance.
>
> **Q1:** Limited Theoretical Discussion: The paper’s justification for DAMamba’s improved spatial modeling remains primarily empirical. While the practical benefits of DAS are evident, the work does not rigorously dissect or quantify how much spatial adjacency (or other higher-order image structure) is preserved versus prior scan methods, beyond anecdotal visualizations and gains in accuracy—this limits mechanistic insight.
>
> **R1:** We appreciate the reviewer’s suggestion for a deeper theoretical exploration. At present, the paper primarily demonstrates the advantages of Dynamic Adaptive Scan (DAS) through experimental accuracy and visualizations. To further enhance the understanding of its mechanism, we will incorporate the following improvements in the final version:
>
> Introduction of quantitative evaluation metrics: We will introduce a metric based on spatial position reconstruction error to quantitatively assess DAS’s ability to preserve the original spatial structure, in comparison with other scanning strategies such as Sweep Scan or Local Scan.
>
> Inter-class comparative analysis: We will analyze the overlap between DAS scanning regions and ground-truth object boundaries across different image categories to quantify how effectively DAS learns to focus on target regions.
>
>
> **Q2:** I believe that DAMamba’s performance may stem from overfitting to a fixed resolution. I’m genuinely curious—what would happen if the authors extended DAMamba to different resolutions? If it fails to maintain strong performance across varying resolutions, then I would consider the overall architectural design to lack sufficient insight.
>
> **R2:** We thank the reviewer for their attention to the generalization ability of our model. In our image classification experiments, we did use a fixed resolution of 224×224 for both training and evaluation, which is the standard practice adopted by most current methods.
>
> However, in downstream vision tasks such as object detection and instance segmentation (using the Mask R-CNN 1× setting), the image resolutions during both training and testing are dynamically varied. Specifically, the shorter side of each image is resized to 800 pixels, while the longer side is capped at 1333 pixels, with the original aspect ratio preserved.
>
> In addition, we also adopted the more advanced Mask R-CNN 3× training strategy in our experiments. This setup introduces stronger data augmentation mechanisms, including multi-scale training. Concretely, during training, each image is randomly resized to one of 11 predefined scales, with the short side ranging from 480 to 800 pixels and the long side limited to a maximum of 1333 pixels. These dynamically varying input resolutions pose a higher challenge to the model's scale robustness.
>
> Under such settings with dynamic input resolutions, our model still achieves performance significantly superior to mainstream baselines in both object detection and instance segmentation tasks. This strongly validates the robust generalization capability of our model when handling vision tasks across varying resolutions.
>
> **Q3:** While major design elements are described, there are some notational ambiguities—e.g., in the formalism in Section 3.2 around offset prediction and feature sampling, more explicit pseudocode or implementation details would aid reproducibility for non-experts.
>
> **R3:** We thank the reviewer for their concern regarding reproducibility. Although we have included implementation code in the supplementary materials, we acknowledge that this may not be sufficient. In the final version, we will provide the following additions:
>
> Pseudocode for the DAS module: We will include PyTorch-style pseudocode that outlines the forward pass of the DAS module, covering key components such as offset computation, coordinate transformation, and bilinear interpolation sampling.
>
> Additional implementation details: We will provide more comprehensive information on aspects such as the initialization of the offset learning range, the gradient flow path, and the interface between DAS and the SSM module.
>
>
> **Q4:** While Figure 5 in the supplement visually contrasts regions of attention, the main methods for inspecting what is actually learned by the scanning process are limited to qualitative renderings, lacking systematic quantitative evaluation (e.g., are certain classes of object consistently mis-scanned?)
>
>
> **R4:** We thank the reviewer for pointing out the limitations in our interpretability analysis. In the current version, we primarily rely on visualizations of DAS scanning paths on ImageNet samples as intuitive evidence. To enhance the systematic nature of our analysis, we plan to include the following additions in the final version:
>
> Category-level statistical analysis: We will compute the overlap between DAS scanning regions and ground-truth objects across different ImageNet classes, analyzing how the model attends to objects of varying shapes and sizes.
>
> Failure case analysis: We will examine samples with degraded performance (e.g., misclassified images) to investigate whether the DAS scanning regions deviate from the key object areas.

---

> > ### Comment · Reviewer_xcKP · 2025-08-08
> >
> > Thank you for the detailed and substantial response. My concerns have been fully addressed. I am positive for accepting this paper.

---

> > > ### Author Response · Authors · 2025-08-08
> > >
> > > We sincerely thank the reviewer for the positive feedback and for considering raising the score.

---

### Official Review · Reviewer_e5U2 · 2025-07-04

**Clarity:** 3
**Significance:** 3
**Originality:** 3
**Rating:** 5
**Confidence:** 3

**Summary:**

This paper proposes a new scanning path for recurrent Mamba-based vision processing. The authors claim that manually designed scans that flatten image patches into sequences disrupt the original semantic spatial adjacency of the image and lack flexibility. To address this, they propose a learnable method termed "dynamic adaptive scan" that adaptively allocates scanning orders and regions. Using this in Vision Mamba (based on a VMamba-like architecture), the authors show improved performance on classification, object detection, and segmentation tasks.

**Questions:**

1) On Line 181, it is mentioned that sample feature vectors of interest are arranged in the order of top to bottom and left to right. This is unclear in the sense that you are doing 4-way scanning, or it's just one scan path with flattening in the mentioned order?

2) The effect of DAScan, Convpos, and ConvFFN is only discussed on the smallest version of the model. It would be helpful to see the effect of all three on at least DAMamba-S or DAMamba-B.

3) Other questions are discussed in the weakness section above.

**Ethical Concerns:**

["NO or VERY MINOR ethics concerns only"]

**Final Justification:**

I thank the authors for the rebuttal. Most of my concerns are addressed. I will raise the score.

**Limitations:**

The limitations description is missing. Authors could talk about the effect of the number of reference points required for the dynamic adaptive scan and what will happen if very few points are used.

**Quality:**

3

**Strengths And Weaknesses:**

Strengths:

1) The proposed offset prediction network, designed to predict offsets for data-driven dynamic adaptive scan, is lightweight and can be easily plugged into any vision framework. It uses a simple depthwise convolution and a linear layer to predict the offsets.

2) The experiment section is exhaustive, covering 4 model sizes across 3 standard datasets and corresponding 4 tasks.

Weaknesses:

While the paper mentions that their major contribution is a dynamic scan path, there is an issue when directly comparing with VMamba or LocalMamba. The DAMamba method diverges from the VMamba in two directions: 1) Number of Channels and Blocks (Table 1), and 2) Use of additional ConvFFM and Convpos. Because of this, it is not clear how much improvement actually comes from the proposed dynamic scan version.

For point 1, since the architecture of DAMamba reduces the channels in the last layer and instead increases the number of blocks, it is not clear whether performance improvement is coming due to that over VMamba and LocalMamba. I strongly suggest authors try to run a DAMamba-S with the same channels ([96,192,384,768]) and blocks ([2,2,27,2]) configuration as VMamba-S, and then show the comparison with VMamba-S and LocalMamba-S. If not possible due to a compute issue, at least DAMamba-T with VMamba-T channel and blocks configuration needs to be conducted.

For point 2, in the current DAMamba-S configurations, I suggest the authors try the following two experiments: 1) Remove the DAScan and just use a simple 4-way VMamba scan while keeping all other components and configurations of the proposed DAMamba-S. 2) Remove DAScan and replace it with LocalVMamba-based 4-way local scanpath while keeping other components consistent. This experiment will clearly show if the dynamic scan is signifcantly effective over vanilla or local scanpath.

If possible time considering, also transfer these models for segmentation tasks and demonstrate the performance of above requested models on that task too.

---

> ### Author Rebuttal · Authors · 2025-07-31
>
> We sincerely appreciate your valuable and encouraging comments on our technical contributions and performance.
>
> **Q1:** On Line 181, it is mentioned that sample feature vectors of interest are arranged in the order of top to bottom and left to right. This is unclear in the sense that you are doing 4-way scanning, or it's just one scan path with flattening in the mentioned order?
>
> **R1:** Thank you to the reviewer for pointing out the ambiguity in terminology. In line 181 of the paper, what we meant is: the sampled patches in DAS are ordered from top to bottom and left to right based on their original positions. This ordering is solely used to construct a one-dimensional sequence for SSM processing, which is scanned only once—not a four-way scan in the traditional sense.
>
> Unlike the 4-way scan used in VMamba, our method employs a single scan path for flattening. However, the patch order is dynamically predicted and non-fixed, and then sorted according to their original coordinates (top to bottom, left to right). This preserves local adjacency and enhances semantic continuity. We will further clarify the definition of the scan path in the final version and emphasize that our method does not use multi-directional scanning.
>
> **Q2:** The effect of DAScan, Convpos, and ConvFFN is only discussed on the smallest version of the model. It would be helpful to see the effect of all three on at least DAMamba-S or DAMamba-B.
>
> **R2:** We agree with the reviewer’s point. Due to limited computational resources and time constraints during the rebuttal period, we will conduct additional ablation studies on DAMamba-S and DAMamba-B in the final version.
>
> **Q3:** Other questions are discussed in the weakness section above.
>
> **R3:** Due to the limited time and computational resources during the rebuttal period, we conducted experiments on DAMamba-T using the same channel and block configurations as VMamba-T. The results are shown below, demonstrating that our model still significantly outperforms both VMamba and LocalMamba:
>
> Method | #Params(M)|FLOPs(G)|Accuracy (%)|
> ------  | ------ |  ------ |------ |
> DAMamba-T	|  26|4.8 |83.8 |
> DAMamba-T(VMamba config)	|  30|5.3 |83.5 |
> VMamba-T | 30 |4.9 |82.6 |
> LocalMamba-T |26|5.7 |82.7 |
>
>
> Due to the same constraints of time and computational resources during the rebuttal, we also performed a comparative study on different scan methods using DAMamba-P. To ensure fairness, we disabled both convpos and convfffn, used DAMamba-P as the baseline, and replaced its scan mechanism with the 4-way VMamba scan and 4-way Local scan. The results, shown below, clearly demonstrate the superiority of our DAS method over the other two scanning strategies.
>
> Method  |Accuracy (%)|
> ------  | ------ |
> Dynamic Adaptive Scan(Ours)	|  78.3|
> Sweeping Scan (VMamba)|  77.7|
> Local Scan (LocalMamba) | 77.9 |

---

> ### Comment · Reviewer_e5U2 · 2025-08-03
>
> I thank the authors for the rebuttal. Most of my concerns are addressed. I will raise the score.
>
> Please report the new results and a more thorough analysis of them for the Small model size in the camera-ready.

---

> > ### Author Response · Authors · 2025-08-03
> >
> > We sincerely thank the reviewer for the positive feedback and for considering raising the score. We appreciate your suggestion and will report the new results along with a more thorough analysis for the Small model size in the camera-ready version.

---

### Note · Authors · 2025-08-13

We sincerely thank the Area Chairs and the four reviewers for their thorough and comprehensive review of our work, as well as for the valuable time and effort they devoted throughout the entire evaluation process. During the rebuttal stage, the reviewers actively engaged in discussions and, based on their full understanding and recognition of our work, raised their scores, for which we are deeply grateful. At the same time, the many insightful comments and constructive suggestions provided by the reviewers not only precisely identified directions for improvement but also offered important guidance for further enhancing the quality and clarity of the paper. We will carefully adopt and implement these suggestions to present a more refined and complete version in the final submission. Once again, we sincerely thank the Area Chairs and the four reviewers for their support, encouragement, and insightful feedback.

---

### Decision · Program_Chairs · 2025-09-17

**Decision:**

Accept (poster)

**Comment:**

The paper proposes a dynamic adaptive scan for Mamba architectures. The approach is evaluated for several computer vision tasks like image classification, object detection, and instance segmentation. The reviewers appreciate the thorough evaluation and the practical value of the proposed approach. The reviewers, however, had also some concerns regarding the comparison of the proposed dynamic scan with other scanning approaches and a potential overfitting of the approach to a fixed resolution. The reviewers also asked to provide more technical details and improve the discussion of related work. The rebuttal resolved the questions regarding the experimental evaluation and overfitting issues. The rebuttal also promised to extend the discussion of related work and provide more technical details. After the rebuttal, the reviewers unanimously recommend the acceptance of the paper.

The AC agrees with the recommendation of the reviewers. While the novelty is limited and the paper needs to discuss the difference to recent related works that use dynamic adaptive scans for Mamba and other network architectures more in detail as raised by Reviewer RoM8, the results in the paper and the additional ablation studies in the rebuttal demonstrate the advantage of the proposed approach compared to other scanning techniques. The ablation studies regarding different scanning techniques need to be included in the paper.